# Accelerating equilibrium spin-glass simulations using quantum annealers via generative deep learning

Giuseppe Scriva[1,2]*, Emanuele Costa[1,2], Benjamin McNaughton[1,3] and Sebastiano Pilati[1,2]

**1** Physics Division, School of Science and Technology, University of Camerino, I-62032 Camerino (MC), Italy
**2** INFN, Sezione di Perugia, I-06123 Perugia, Italy
**3** Department of Physics, University of Antwerp, B-2020 Antwerp, Belgium
* giuseppe.scriva@unicam.it

December 15, 2022

## Abstract

**Adiabatic quantum computers, such as the quantum annealers commercialized by D-Wave Systems Inc., are routinely used to tackle combinatorial optimization problems. In this article, we show how to exploit them to accelerate equilibrium Markov chain Monte Carlo simulations of computationally challenging spin-glass models at low but finite temperatures. This is achieved by training generative neural networks on data produced by a D-Wave quantum annealer, and then using them to generate smart proposals for the Metropolis-Hastings algorithm. In particular, we explore hybrid schemes by combining single spin-flip and neural proposals, as well as D-Wave and classical Monte Carlo training data. The hybrid algorithm outperforms the single spin-flip Metropolis-Hastings algorithm. It is competitive with parallel tempering in terms of correlation times, with the significant benefit of a much shorter equilibration time.**

# 1  Introduction

Simulating the low-temperature equilibrium properties of frustrated, disordered Ising models is a hard computational task for classical computers. It plays a central role in the understanding of glasses [1–3], and it is also connected to relevant quadratic binary optimization problems, whose solution (in the absence of constraints) corresponds to the identification of the spin configuration(s) with the lowest energy [4]. Markov chain Monte Carlo (MC) simulations driven by simple implementations of the Metropolis-Hastings (MH) algorithm [5, 6] are affected by diverging correlation times at low temperatures [7]. Various smart sampling schemes have been developed; arguably, the most relevant are parallel tempering (PT) [8] and the isoenergetic cluster updates [9, 10]. Anyway, the research for further developments is still vivid [11].

In recent years, machine learning (ML) techniques have been widely adopted in computational physics [12–14]. In particular, generative deep learning has proven promising for accelerating stochastic simulations, addressing challenging multimodal molecular systems [15–17], lattice models [18, 19], ferromagnetic and random spin models [20–22, 22–24], solid-state systems [25], as well as quantum models [26–31]. If appropriately trained, generative neural networks (NNs) are able to generate particularly efficient MC updates. However, it was noted that the training based on the reverse Kullback-Leibler divergence is susceptible to mode collapse problems [15, 17, 32–34]. On the other hand, the unsupervised learning – based on the forward Kullback-Leibler divergence minimization – is also possible, but it requires training datasets produced either from previous simulations or from experiments. Simulated data might be produced, e.g., via sequential tempering [23], but this might involve an uncontrolled computational cost [34]. This encourages one to explore the experimental route. Interestingly, it has recently been proven that ML algorithms trained on data produced by quantum experiments are, in theory, able to solve otherwise classically intractable computational tasks [35].

Very recently, a quantum algorithm designed to sample from the Boltzmann distribution of Ising models has been presented [36,37]. It exploits universal gate-based quantum computers. While steadily growing, the size of these devices is still too small to clearly observe diverging correlation times in spin-glass models. On the other hand, the quantum annealers (QAs) commercialized by D-Wave Systems already feature thousands of qubits (see, e.g., Refs. [38,39]). They are routinely used to tackle optimization problems, and they have also been adopted to train neural networks [40–43]. Notably, in a recent study they were used to sample rare transitions in challenging molecular systems [44], but the employed approach required inferring proposal probabilities via frequency histograms.

In this article, we show how to combine generative deep learning and QAs to improve thermodynamic-equilibrium simulations of spin glasses. Autoregressive NNs are trained on spin configurations produced by a D-Wave Advantage QA, and then used to generate smart proposals for the MH algorithm. The autoregressive property provides the exact proposal probabilities required to compute the MH acceptance [21, 23], thus avoiding frequency histograms.

The testbed models we consider are sizable Ising models on square lattices with nearest neighbor and also next-nearest neighbor frustrated random interactions. We implement neural MC updates, as well as hybrid sampling schemes which combine neural updates with standard single spin-flip (SSF) updates. This eliminates possible ergodicity breakdowns due to configuration-space regions not accessible by the D-Wave samples. The augmentation of D-Wave configurations with classical MC data is also investigated, as well as the role of different annealing times. We benchmark the hybrid MC scheme against the SSF algorithm and the powerful PT method. In the challenging low-temperature regime, the hybrid scheme outperforms the SSF algorithm in terms of correlation times, and it is competitive with PT, with the significant benefit of a reduced equilibration time. While our method does not require the D-Wave spin configurations to exactly mimic the Boltzmann distribution, our findings indicate that they are sufficiently representative of the relevant low-energy sectors to strongly boost low-temperature equilibrium simulations.

The article is organized as follows: Section 2 introduces the Ising models we consider. Section 3 describes the SSF-MC algorithm, as well as the neural (N-MC) and the hybrid MC algorithms (H-MC). It also provides some details on our PT simulations. Section 4 introduces the autoregressive neural networks and their training protocol. Section 5 provides some details on the quantum annealing protocols performed on the D-Wave Advantage QA and it describes the sampled configurations. Additional details on the embedding of our lattice setups on Advantage's native graph are provided in Appendix A. Appendix B describes the QA's runtime utilization. In Section 6 we analyze the performances of the N-MC and of the H-MC algorithms on sizable instances of the adopted spin-glass models. Comparison is made against the SSF-MC and the PT algorithms. Our main findings are summarized in Section 7, with some comments on future perspectives.

## 2   Random Ising Hamiltonians

We consider spin-glass models [45] defined on two-dimensional square lattices. The Hamiltonian reads

$$H(\boldsymbol{\sigma}) := \sum_{\langle i,j \rangle} J_{ij} \sigma_i \sigma_j, \tag{1}$$

where $\sigma_i \in \{\pm 1\}$ are binary spin variables at the sites $i = 1, \ldots, N$, $\boldsymbol{\sigma} = (\sigma_1, \ldots, \sigma_N)$ indicates the whole spin configuration, and $N$ is the total number of spins. $J_{ij}$ is the coupling between spins $i$ and $j$. The symbol $\langle i, j \rangle$ indicates that the sum is restricted to nearest-neighbor or up next nearest-neighbor spins, as detailed below. Open boundary conditions are assumed. The Boltzmann distribution is defined as

$$h(\boldsymbol{\sigma}) := \exp[-\beta H(\boldsymbol{\sigma})]/Z, \tag{2}$$

where $\beta = 1/(k_B T)$ is the (rescaled) inverse temperature, T is the temperature, and the normalization term $Z := \sum_{\boldsymbol{\sigma}} \exp[-\beta H(\boldsymbol{\sigma})]$ is the partition function. Throughout the article, the energy units are set so that the Boltzmann constant is $k_B = 1$. We are interested in the thermodynamic properties, such as the average energy per spin $E/N := \langle H(\boldsymbol{\sigma}) \rangle/N$, where the brackets indicate the expectation value over the Boltzmann distribution.

In the following, three lattice setups will be addressed as a testbed for our methods: (i) a square lattice with $N = 100$ spins and only nearest neighbor interaction. The couplings $J_{ij}$ are sampled from a Gaussian distribution with zero mean and unit variance, namely, $\mathcal{N}(0, 1)$. The corresponding coordination number for internal spins is $z = 4$. (ii) A square lattice with $N = 484$ spins and only nearest-neighbor interaction; here, the couplings are sampled from a uniform distribution in the range $J_{ij} \in [-1, 1]$, namely, Unif$[-1, 1]$. This model will be

referred to as the $N = 484 \, (z = 4)$ setup. (iii) A square lattice with $N = 484$ spins, including both nearest-neighbor and next-nearest neighbor couplings on the diagonal, corresponding to $z = 8$ for internal spins. All couplings are sampled from Unif$[-1, 1]$. We refer to this model also as the $N = 484 \, (z = 8)$ setup.

The three setups present different levels of difficulty for computational algorithms. Indeed, the ground-state configurations of square lattices with only nearest-neighbor interactions can be identified with exact algorithms. Furthermore, while SSF-MC simulations are affected by long correlation time in the regime $\beta \simeq 1$ [46], this model hosts a spin-glass phase with finite Edward-Anderson order parameter only in the zero-temperature limit [47, 48]. The inclusion of next-nearest neighbor interactions leads to a non-planar topology. In this case, exactly identifying the ground state is, in general, not possible with polynomial-time algorithms [49].

In Sections 4, 5 and 6, the $N = 100$ setup (i) is used to illustrate the behavior of the methods described in Section 3. The $N = 484 \, (z = 4)$ setup (ii) allows demonstrating that our H-MC method outperforms the SSF-MC algorithm. In the $N = 484 \, (z = 8)$ setup (iii), the SSF-MC algorithm becomes impractical, and we compare the H-MC method against the powerful PT technique.

# 3 Markov chain Monte Carlo algorithms

## 3.1 Single-spin flip Monte Carlo algorithm

MC simulations allow accurately estimating thermodynamic expectation values by sampling spin configurations according to the Boltzmann distribution Eq. (2) [5]. Starting from an arbitrary (e.g., random) configuration, random updates from a configurations $\boldsymbol{\sigma}$ to another one $\boldsymbol{\sigma}'$ are generated according to a transition probability $P(\boldsymbol{\sigma}'|\boldsymbol{\sigma})$. Provided the Markov chain is irreducible and aperiodic [50], a sufficient condition to ensure convergence to the target stationary distribution, in our case the Boltzmann distribution $h(\boldsymbol{\sigma})$, is represented by the detailed balance condition

$$P(\boldsymbol{\sigma}'|\boldsymbol{\sigma})h(\boldsymbol{\sigma}) = P(\boldsymbol{\sigma}|\boldsymbol{\sigma}')h(\boldsymbol{\sigma}'), \tag{3}$$

for all $\boldsymbol{\sigma}$ and $\boldsymbol{\sigma}'$ [51]. A convenient criterion to satisfy Eq. (3) is to decompose the transition probability $P(\boldsymbol{\sigma}'|\boldsymbol{\sigma})$ using a non-negative column-normalized proposal distribution $Q(\boldsymbol{\sigma}'|\boldsymbol{\sigma})$ and a suitable acceptance probability $A(\boldsymbol{\sigma}'|\boldsymbol{\sigma})$; one obtains

$$P(\boldsymbol{\sigma}'|\boldsymbol{\sigma}) = \begin{cases} Q(\boldsymbol{\sigma}'|\boldsymbol{\sigma})A(\boldsymbol{\sigma}'|\boldsymbol{\sigma}) & \text{if } \boldsymbol{\sigma} \neq \boldsymbol{\sigma}', \\ 1 - \sum_{\boldsymbol{\sigma}'' \neq \boldsymbol{\sigma}} Q(\boldsymbol{\sigma}''|\boldsymbol{\sigma})A(\boldsymbol{\sigma}''|\boldsymbol{\sigma}) & \text{if } \boldsymbol{\sigma} = \boldsymbol{\sigma}'. \end{cases} \tag{4}$$

An efficient and popular choice for the acceptance probability, which satisfies Eq. (3), is the following [5, 6]

$$A(\boldsymbol{\sigma}'|\boldsymbol{\sigma}) := \min\left(1, \frac{h(\boldsymbol{\sigma}')Q(\boldsymbol{\sigma}|\boldsymbol{\sigma}')}{h(\boldsymbol{\sigma})Q(\boldsymbol{\sigma}'|\boldsymbol{\sigma})}\right). \tag{5}$$

Importantly, since only ratios of Boltzmann-distribution values are used, the (intractable) computation of the partition function $Z$ is not required. Moreover, one notices that Eq. (5) simplifies for symmetric proposals, i.e, such that $Q(\boldsymbol{\sigma}'|\boldsymbol{\sigma}) = Q(\boldsymbol{\sigma}|\boldsymbol{\sigma}')$ for any $\boldsymbol{\sigma}$ and $\boldsymbol{\sigma}'$. A common choice for the proposal distribution is the SSF algorithm, whereby the flipping of a randomly selected spin is proposed. This corresponds to $Q(\boldsymbol{\sigma}'|\boldsymbol{\sigma}) = 1/N$ if $\boldsymbol{\sigma}'$ and $\boldsymbol{\sigma}$ differ for one (and only one) spin, while $Q(\boldsymbol{\sigma}'|\boldsymbol{\sigma}) = 0$ otherwise. While this simple algorithm is suitable for quite variegate physical systems, it is known to suffer from diverging correlation times close to phase transitions or in glassy phases, effectively breaking ergodicity in feasible simulation

times [7]. In Section 6, the SSF simulation times $\tau$, representing the number of sweeps, will be compared to other algorithms. For the SSF algorithm, a sweep corresponds to $N$ spin-flip attempts. This definition follows a standard convention adopted in the literature.

## 3.2   Neural Monte Carlo algorithm

To improve beyond the SSF algorithm, smarter proposal distributions $Q(\boldsymbol{\sigma}'|\boldsymbol{\sigma})$ need to be implemented. Recent studies proposed using generative NN, specifically, auto-normalizing flows or autoregressive models. These assign a properly normalized probability (or probability density, in the case of continuous variables) to each system configuration. We indicate this probability as $q(\boldsymbol{\sigma})$. Furthermore, they allow efficient direct sampling of this probability distribution, without invoking a Markov process. Henceforth, one sets [21, 23]

$$Q(\boldsymbol{\sigma}'|\boldsymbol{\sigma}) = q(\boldsymbol{\sigma}'). \tag{6}$$

Formally, convergence to the target distribution is guaranteed as long as $q(\boldsymbol{\sigma}) > 0$ for all configurations $\boldsymbol{\sigma}$ such that $h(\boldsymbol{\sigma}) > 0$. This condition is automatically fulfilled for the autoregressive network described in Section 4, since one has $q(\boldsymbol{\sigma}) > 0$ for any $\boldsymbol{\sigma}$, due to our choice of output activation function in Eq. (10). In practice, however, $q(\boldsymbol{\sigma})$ might be exponentially small for configurations where the Boltzmann weight is sizable. This would lead to an effective ergodicity breakdown in feasible simulation times. On the other hand, if the network learns a good approximation of the Boltzmann distribution, i.e., if $q(\boldsymbol{\sigma}) \sim h(\boldsymbol{\sigma})$ for all $\boldsymbol{\sigma}$, the acceptance probability is $A(\boldsymbol{\sigma}'|\boldsymbol{\sigma}) \simeq 1$, leading to an efficient ergodic simulation. This algorithm is referred to as neural MC (N-MC), and it is detailed in the Algorithm 1.

---

**Algorithm 1** Neural Monte Carlo

---

**Require:** $\tau$, NN()                                                                    ▷ Sweeps and trained NN
  $\boldsymbol{\sigma}, q(\boldsymbol{\sigma}) \leftarrow$ NN()                                    ▷ Sample and its probability
  $i \leftarrow 1$
  **for** $i \leq \tau$ **do**
      $\boldsymbol{\sigma}', q(\boldsymbol{\sigma}') \leftarrow$ NN()
      $r \leftarrow \frac{h(\boldsymbol{\sigma}')}{h(\boldsymbol{\sigma})} \cdot \frac{q(\boldsymbol{\sigma})}{q(\boldsymbol{\sigma}')}$
      $A \leftarrow \min(1, r)$                                                         ▷ Acceptance probability
      **if** $A >$ Unif$[0, 1)$ **then**
          $\boldsymbol{\sigma}, q(\boldsymbol{\sigma}) \leftarrow \boldsymbol{\sigma}', q(\boldsymbol{\sigma}')$
      **end if**
      $i \leftarrow i + 1$
  **end for**

---

We point out that the computational cost of neural proposal generation can be off-loaded and executed by exploiting graphical processing units (GPUs). Each N-MC update requires the computation of the whole configuration energy. This is comparable to $N$ SSF updates, assuming that in a single update only the energy difference is computed. Thus, for the N-MC algorithm, we define a sweep as proposing, and then accepting or rejecting, one system configuration.

## 3.3   Hybrid Monte Carlo algorithm

When the generative NN does not efficiently sample all physically relevant spin configurations, i.e., those corresponding to sizable values of the Boltzmann weight, the N-MC algorithm becomes pathologically inefficient. The expectation values estimated in feasible simulation times

might be biased. This problem might be remediated via a hybrid MC (H-MC) scheme which (sequentially) combines SSF-MC and N-MC updates[1]. The sequence satisfies the detailed balance condition since the individual updates do. Specifically, we implement $N$ SSF updates, (deterministically) followed by one N-MC update. The whole sequence will be referred to as one sweep. Its computational cost is of the same order as one sweep of the SSF-MC or the N-MC algorithms. The H-MC scheme is detailed in Algorithm 2. It aims at eliminating the drawbacks of both the SSF-MC and the N-MC algorithms, combining their functionalities. The H-MC updates are supposed to perform large leaps between distance configurations (in terms of Hamming distance). The SSF moves allow exploring the neighborhoods around the configurations reached by the leaps, allowing exploring regions that cannot be sampled by the NN.

The inefficiency of the N-MC algorithm is expected to originate from the possible bias of the configuration dataset used to train the generative NN. As discussed in Section 6, this problem sometimes occurs with the configurations generated by a D-Wave QA. This device is designed to sample low-energy configurations. Therefore, the trained NN will not sample high-energy configurations, which are relevant at high temperatures. Beyond the H-MC scheme, an alternative (possibly complementary) strategy consists in using hybrid datasets, including both configurations generated by a D-Wave device and by SSF-MC simulations performed in the feasible regime, namely, high or intermediate temperatures. Results obtained with this additional protocol are discussed in Section 6.

---

**Algorithm 2** Hybrid Monte Carlo

---

**Require:** $\tau, N$, NN()                                                    ▷ Sweeps, spins and NN
  $\boldsymbol{\sigma}, q(\boldsymbol{\sigma}) \leftarrow$ NN()                                          ▷ Sample and its probability
  $i \leftarrow 1$
  **for** $i \leq \tau \cdot N + \tau$ **do**                                   ▷ A step is a SSF sweep plus a NN proposal
    **if** mod $(i, N+1) \neq 0$ **then**                                       ▷ Attempt $N$ spin flips
      $k \leftarrow$ Unif$\{1, N\}$                                            ▷ Pick a spin to flip
      $\boldsymbol{\sigma}' \leftarrow (\sigma_1, \cdots, -\sigma_k, \cdots, \sigma_N)$
      $r \leftarrow h(\boldsymbol{\sigma}')/h(\boldsymbol{\sigma})$
      $q(\boldsymbol{\sigma}') \leftarrow$ NN$(\boldsymbol{\sigma}')$                                   ▷ Compute $q(\boldsymbol{\sigma}')$
    **else**                                                                    ▷ Attempt one neural step
      $\boldsymbol{\sigma}', q(\boldsymbol{\sigma}') \leftarrow$ NN()
      $r \leftarrow \frac{h(\boldsymbol{\sigma}')}{h(\boldsymbol{\sigma})} \cdot \frac{q(\boldsymbol{\sigma})}{q(\boldsymbol{\sigma}')}$
    **end if**
    $A \leftarrow \min(1, r)$
    **if** $A > $ Unif$[0, 1)$ **then**
      $\boldsymbol{\sigma}, q(\boldsymbol{\sigma}) \leftarrow \boldsymbol{\sigma}', q(\boldsymbol{\sigma}')$
    **end if**
    $i \leftarrow i + 1$
  **end for**

---

## 3.4 Parallel Tempering Monte Carlo algorithm

The parallel tempering (PT) method [8, 52], also known as exchange Monte Carlo method, represents one of the most suitable algorithms to simulate frustrated spin models in the low-temperature regime. It allows overcoming free energy barriers that separate metastable states,

---

[1]A parallel stochastic combination of neural and SSF updates is also possible, but it requires a modified acceptance probability. Since it does not lead to efficiency improvements in our benchmarks, we do not discuss it further

thus performing ergodic simulations even when two or many metastable states compete. It is employed in Section 6 to simulate the challenging lattice $N = 484$ $(z = 8)$ setup, for which the SSF-MC algorithm is impractical. It constitutes a relevant performance benchmark for the N-MC and the H-MC algorithms.

The PT method is based on $M$ non-interacting replicas of the system, each associated to a distinct inverse temperature $\beta_m$, with $m = 1, \ldots, M$, such that $\beta_m < \beta_{m+1}$. The spin configurations of each replica are sampled from the Boltzmann distribution $h_m(\boldsymbol{\sigma})$ at the corresponding $\beta_m$. This is achieved with standard SSF-MC updates. Additionally, one introduces swap updates that attempt to exchange the configurations $\boldsymbol{\sigma}_m$ and $\boldsymbol{\sigma}_{m+1}$ associated to two adjacent replicas. The corresponding acceptance probability is

$$A_s(\boldsymbol{\sigma}_m, \beta_m | \boldsymbol{\sigma}_{m+1}, \beta_{m+1}) := \min\left(1, \exp(\Delta)\right), \tag{7}$$

where $\Delta = (\beta_{m+1} - \beta_m)(H(\boldsymbol{\sigma}_{m+1}) - H(\boldsymbol{\sigma}_m))$. The detailed balance equation is satisfied if the swaps are proposed independently on the current state [53].

The number of replicas $M$ required for an efficient simulation is known to scale as $\sqrt{N}$ [54]. Choosing the inverse temperatures $\beta_m$ is not straightforward. A reasonable ex-ante criterion is to fix all ratios $\beta_{m+1}/\beta_m$ to the same constant. This is determined by the smallest inverse temperatures $\beta_1$, by the largest one $\beta_M$, and by the chosen number of replicas $M$. $\beta_1$ shall be small enough to allow an efficient ergodic SSF-MC simulation. $\beta_M$ is chosen according to the lowest temperature regime of interest. We adopt this criterion in the comparison of correlation times in Section 6, setting $\beta_1 = 0.01$, $\beta_M = 10$, and $M = 22$. Alternatively, the inverse temperatures can be chosen so that all average swap acceptance rates are close to, e.g., 20%. This is a time-consuming procedure, requiring an ex-post parameter optimization. We adopt this criterion to obtain highly accurate energy expectation values for precise benchmarking. In this case, we set $\beta_1 = 0.1$, $\beta_M = 10$, and $M = 40$.

Due to the use of replicas, the PT algorithm implies a significant overall computational overhead compared to the SSF-MC simulations. However, the replicas can be executed in parallel using different computing cores, and they simultaneously provide information on different temperatures. Furthermore, the cost of swap updates, which is, in practice, mostly determined by inter-process communications, might be suppressed via an efficient implementation of inter-process communication. For this, we follow the implementation of Ref. [55]. Therefore, when comparing the PT performance with other algorithms, we define a PT sweep as $N$ SSF updates per replica and one swap update per pair of adjacent replicas. This choice is favorable to the PT algorithm, and it is intended to implement a stringent benchmark for the other MC algorithms.

# 4 Autoregressive neural networks

Generative neural networks allow inferring an unknown probability distribution $p(\boldsymbol{x})$ from a set of $T$ samples $\{\boldsymbol{x}^{(t)}\}_{t=1}^T$ [56]. Here, we consider $N$-dimensional arrays $\boldsymbol{x} = (x_1, \ldots, x_N)$, with $x_i \in \{0, 1\}$. These can be associated to spin configurations $\boldsymbol{\sigma}$, with $\sigma_i \in \{\pm 1\}$, via the invertible map $\boldsymbol{x} = (\boldsymbol{\sigma} + 1)/2$. For some of the NNs discussed hereafter, the input has to be a one-dimensional vector. In that case, we flatten the two-dimensional lattice in the row by row order.

In the N-MC and the H-MC methods of Section 3, the generative NN is used to generate smart proposals. The NN is required to assign a properly normalized probability to each configuration, and to allow efficient direct sampling. For this task, recent studies employed either auto-normalizing flows [17, 24, 25, 33], in the case of continuous-variable problems, or autoregressive NNs, in the case of spin models. With the autoregressive property, the learned

probability distribution is written as a product of chained conditional distributions, in the form

$$p(\boldsymbol{x}) = \prod_{i=1}^{N} p(x_i \mid \boldsymbol{x}_{<i}), \tag{8}$$

where $\boldsymbol{x}_{<i} = (x_1, x_2, \dots, x_{i-1})$ is a vector with the first $i-1$ elements of the input. Configurations can be efficiently generated via ancestral sampling: after $i-1$ binary variables have been sampled, one sets $x_i = 1$ with (conditional) probability $p(x_i \mid \boldsymbol{x}_{<i})$, and $x_i = 0$ with probability $1 - p(x_i \mid \boldsymbol{x}_{<i})$.

We consider three autoregressive NNs borrowed from the literature, namely, the neural autoregressive distribution estimator (NADE) [57], the masked autoregressive density estimator (MADE) [58], and the so-called PixelCNN [59]. We train them on datasets of spin configurations produced by a D-Wave QA. It is found that MADE outperforms NADE in terms of computational efficiency, both in the training and in the generation phase. Furthermore, MADE reproduces our training datasets (see Section 5) more accurately than PixelCNN. This phenomenon is visualized in the histogram of sampled configuration energies of Fig. 1. One notices that PixelCNN oversamples high-energy configurations. It is worth mentioning that

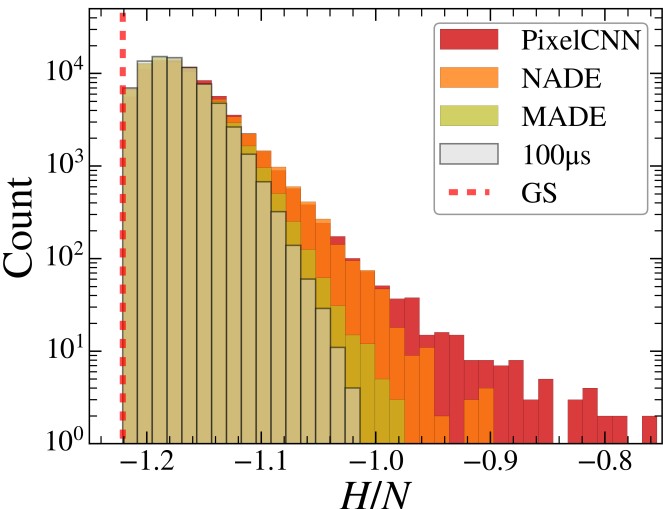

Figure 1: Histograms of $10^5$ configuration energies per spin $H/N$, for a $N = 100$ square lattice with nearest-neighbor couplings. The samples of a D-Wave QA with annealing time $t_a = 100\mu$s (grey) are compared with those of three autoregressive neural networks, namely, NADE, MADE, and PixelCNN. These are trained on the $10^5$ D-Wave configurations. The vertical (red) line corresponds to the ground-state energy, computed using the McGroundstate solver [60].

PixelCNN was recently been adopted to describe clean ferromagnetic Ising models [21]; we attribute the inferior performance found here compared to MADE to our choice of random couplings.

Henceforth, hereafter we illustrate only the architecture of MADE. It is based on an autoencoder [56] composed of an input, a hidden, and an output layer with dense connectivity. Its aim is to obtain a $M$-dimensional hidden representation $f(\boldsymbol{x}) \in \mathbb{R}^M$ of the input $\boldsymbol{x}$, where $M$ also corresponds to the number of neurons in the hidden layer, such that the ($N$-dimensional) reconstruction $\hat{\boldsymbol{x}}$ is as close as possible to $\boldsymbol{x}$. Formally, for a standard autoencoder, one has

$$f(\boldsymbol{x}) = g(\boldsymbol{b} + \boldsymbol{W}\boldsymbol{x}), \tag{9}$$
$$\hat{\boldsymbol{x}} = s(\boldsymbol{c} + \boldsymbol{V}f(\boldsymbol{x})), \tag{10}$$

| Model | Input size $N$ | Hidden size $M$ | Activation | Optimizer | lr | Dataset size T | Batch | Epochs |
|---|---|---|---|---|---|---|---|---|
| MADE 100 | 100 | 512 | LeakyReLU | Adam | $5 \cdot 10^{-3}$ | $10^5$ | 100 | 10 |
| MADE 484 | 484 | 4096 | LeakyReLU | Adam | $5.42 \cdot 10^{-4}$ | $4 \cdot 10^5$ | 96 | 30 |

Table 1: Architecture and hyperparameters of the two autoregressive NNs used for the $N = 100$ lattice (MADE 100) and for the two $N = 484$ ($z = 4$ and $z = 8$) setups (MADE 484).

where $\boldsymbol{W} \in \mathbb{R}^{M \times N}$, $\boldsymbol{V} \in \mathbb{R}^{N \times M}$, $\boldsymbol{b} \in \mathbb{R}^M$ and $\boldsymbol{c} \in \mathbb{R}^N$ are trainable weights and biases, and $g(\boldsymbol{x})$ and $s(\boldsymbol{x})$ are proper activations functions; we adopt the LeakyRelu [61] and the Sigmoid function [56], respectively. MADE is trained via unsupervised learning by minimizing the ensemble binary cross-entropy loss function. For one configuration $\boldsymbol{x}$, this is defined as

$$\ell(\boldsymbol{x}) := \sum_{i=1}^{N} [-x_i \log(\hat{x}_i) - (1 - x_i) \log(1 - \hat{x}_i)]. \tag{11}$$

The weights and biases are optimized via a modified version of stochastic gradient descent, named ADAM [62]. See also Table 1 for technical details. Notice that $\hat{x}_i$ must represent the conditional probability $p(x_i = 1 \mid \boldsymbol{x}_{<i})$. Thus, the loss function also corresponds to the negative log-likelihood

$$
\begin{aligned}
-\log p(\boldsymbol{x}) &= \sum_{i=1}^{N} -\log p(x_i \mid \boldsymbol{x}_{<i}) \\
&= \sum_{i=1}^{N} -x_i \log p(x_i = 1 \mid \boldsymbol{x}_{<i}) - (1 - x_i) \log p(x_i = 0 \mid \boldsymbol{x}_{<i}) \\
&= \ell(\boldsymbol{x}).
\end{aligned}
\tag{12}
$$

To ensure the autoregressive property, two mask matrices $\boldsymbol{M}^W$ and $\boldsymbol{M}^V$ are introduced. They are used to eliminate the connections with previous spins in the chosen (raw by raw) order. Thus, for the autoregressive autoencoder, one has

$$f(\boldsymbol{x}) = g\left(\boldsymbol{b} + \left(\boldsymbol{W} \cdot \boldsymbol{M}^W\right)\boldsymbol{x}\right), \qquad \hat{\boldsymbol{x}} = s\left(\boldsymbol{c} + \left(\boldsymbol{V} \cdot \boldsymbol{M}^V\right)f(\boldsymbol{x})\right), \tag{13}$$

where $\cdot$ indicates here the element-wise product. The masks $\boldsymbol{M}^W$ and $\boldsymbol{M}^V$ are defined so that the product $\boldsymbol{M}^W \boldsymbol{M}^V$ is strictly lower diagonal. We refer the readers to Ref. [58] for the details on this definition. In principle, one can sample an ensemble of masks fulfilling this property; however, our tests show no benefit from considering more than one.

All the NNs are implemented in Lightning [63], a PyTorch [64] research framework, and executed on a NVIDIA RTX A6000 GPU. The most relevant hyperparameters are shown in Tab. 1; some of them are obtained via the Optuna framework [65]. As common in deep learning studies, we split each dataset into training and validation sets, with a $80 : 20$ ratio. The MADE is then trained up to 10 or 30 epochs, using an early stopping criterion via the validation loss function. MADE quickly learns to closely reproduce the energy distribution of D-Wave samples. This allows us, e.g., to characterize the role of different annealing times in N-MC simulations. On the other hand, exactly mimicking the training samples is not essential for the functioning of the N-MC and the H-MC simulations. This means that the training times could be shortened, and one could adopt MADEs with fewer hidden neurons. In our implementation, the training of the largest MADE takes approximately 10s per epoch.

As already mentioned, the proposal configurations can be generated independently of the N-MC and H-MC simulations. This generation can efficiently exploit the massing parallelism of

modern GPUs. With our platform, generating $10^5$ configurations requires about one minute for $N = 100$, and around two minutes and a half for $N = 484$. Notice that a novel configuration must be used in each MC-attempted update. This means that the neural proposals, adopted in the N-MC and the H-MC algorithms, do not constitute a critical computational overhead.

## 5 Configurations from D-Wave quantum annealers

We generate low-energy spin configurations of the Hamiltonian (1) using a quantum annealer (QA) [66, 67] powered by D-Wave Systems. It is equipped with the Advantage processor, featuring more than 5000 programmable qubits. The allowed couplings form the so-called Pegasus graph [39]. The annealing process is described by the following time-dependent Hamiltonian

$$\hat{H} := -\frac{A(s)}{2}\hat{H}_{\text{init}} + \frac{B(s)}{2}\hat{H}_{\text{final}}, \tag{14}$$

where

$$\hat{H}_{\text{init}} := \sum_i \hat{\sigma}_i^x, \qquad \hat{H}_{\text{final}} := \sum_i \tilde{h}_i \hat{\sigma}_i^z + \sum_{i>j} \tilde{J}_{ij} \hat{\sigma}_i^z \hat{\sigma}_j^z. \tag{15}$$

In the above equations, $\hat{\sigma}_i^x$ and $\hat{\sigma}_i^z$ are standard Pauli matrices operating on the qubit $i$, $\tilde{h}_i$ and $\tilde{J}_{ij}$ are the longitudinal fields and the coupling strengths, respectively, $s := t/t_a \in [0, 1]$ is a dimensionless time normalized with the annealing time $t_a$, the function $A(s)$ tunes the intensity of the transverse field operators that form the initial Hamiltonian $\hat{H}_{\text{init}}$, while the function $B(s)$ tunes the scale of problem Hamiltonian $\hat{H}_{\text{final}}$. The latter encodes the classical Hamiltonian (1), corresponding to the optimization problem to be solved.

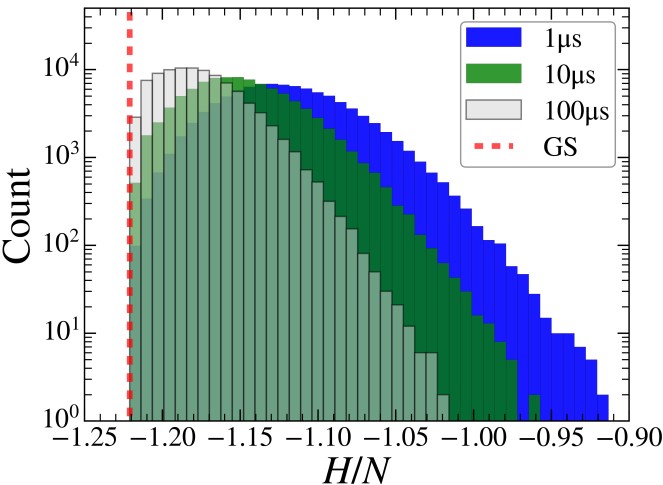

Figure 2: Histograms of $10^5$ configuration energies per spin $H/N$, for the $N = 100$ lattice with nearest neighbor couplings. The three datasets correspond to three annealing times $t_a$. The vertical (red) line indicates the ground-state energy.

The lattice setups defined in Sections 2 are mapped to the Pegasus graph using the native heuristic embedding algorithm of the D-Wave interface. This embedding provides the actual couplings $\tilde{J}_{ij}$ (and eventually, longitudinal fields $\tilde{h}_i$). In this embedding, (short) qubit chains are often used to represent logical spins. The most relevant details of the mapping procedure are provided in Appendix A. Chiefly, we describe the role of the intra-chain coupling strength on the configuration energies of the generated configurations. It is found that, in some cases, appropriately tuning this coupling strength allows reaching significantly lower energies.

| $N$ | $t_a$ | $E_{\mathrm{avg}}/N$ | $E_{\mathrm{min}}/N$ | $E_{\mathrm{gs}}/N$ |
|---|---|---|---|---|
| 100 | 1μs | -1.1191 (1) | -1.22104 | -1.22104 |
| " | 10μs | -1.1474 (1) | -1.22104 | " |
| " | 100μs | -1.17513 (8) | -1.22104 | " |
| 484 (4) | 1μs | -0.70212 (2) | -0.74331 | -0.75503 |
| " | 10μs | -0.72117 (1) | -0.75119 | " |
| " | 100μs | -0.73208 (1) | -0.75347 | " |
| 484 (8) | 1μs | -1.04753 (2) | -1.09698 | -1.09819 |
| " | 10μs | -1.06751 (2) | -1.09709 | " |
| " | 100μs | -1.07829 (1) | -1.09816 | " |

Table 2: Description of the configuration energies per spin $H/N$ sampled by a D-Wave QA, for our three lattice setups. The average $E_{\mathrm{avg}}/N$ and the minimum $E_{\mathrm{min}}/N$ energies per spin are reported for three annealing times $t_a$. Sets of $10^5$ or $4 \cdot 10^5$ samples are considered, for $N = 100$ and $N = 484$, respectively. The ground-state energy $E_{\mathrm{gs}}$ is exactly computed by the McGroundstate solver [60]. The QA finds it only for the $N = 100$ lattice.

The annealing time $t_a$ can be set by the user in the range $[1, 2000]$μs. As reported in Appendix B, the total amount of time required by the D-Wave system is greater than the annealing time alone. The tuning functions are such that $A(0) = 1$ and $B(0) = 0$, so that the initial state is dominated by the transfer fields. One also has $A(1) = 0$ and $B(1) = 1$. This means that, in the absence of decoherence and diabatic transitions, the final state corresponds to a ground-state configuration of the Hamiltonian (1). Assuming coherent annealing, adiabaticity is expected if the annealing times are allowed to increase with the smallest gap $\Delta_{\mathrm{min}}$ between adiabatic ground and first excited states, as: $t_a \sim \Delta_{\mathrm{min}}^{-2}$. Short annealing times and/or decoherence favor diabatic transitions, meaning that higher energy configurations are sampled. This effect is analyzed in the energy histogram in Fig. 2, for the lattice setup $N = 100$ (see definition in Section 2). Additional characteristics of the sampled energies are reported in Table 2. As expected, longer annealing times allow more frequent sampling of low energy configurations, in fact quite close to the ground-state energy.

The ground-state energy is determined using the McGroundstate solver [60], which requires feasible computational times for our lattice setups. Notably, only for the setup with $N = 100$ spins the ground-state energy is exactly met at least once among $10^5$ samples. For the setups $N = 484\,(z = 4)$ and $N = 484\,(z = 8)$, the lowest sampled energy is slightly higher than the ground state. This can be attributed to the smaller energy gaps occurring in larger lattices.

# 6   Results

Here we analyze the efficiency of the N-MC and of the H-MC simulations driven by generative NNs, specifically by MADEs, trained on spin configurations generated by a D-Wave QA. Three testbeds are considered, corresponding to the three lattice setups described in Section 2. They are referred to as $N = 100$, $N = 484\,(z = 4)$, and $N = 484\,(z = 8)$ lattices. Comparisons are made against conventional SSF-MC simulations and more competitive PT simulations. To quantify the algorithmic performances, we consider the configuration-energy auto-correlation

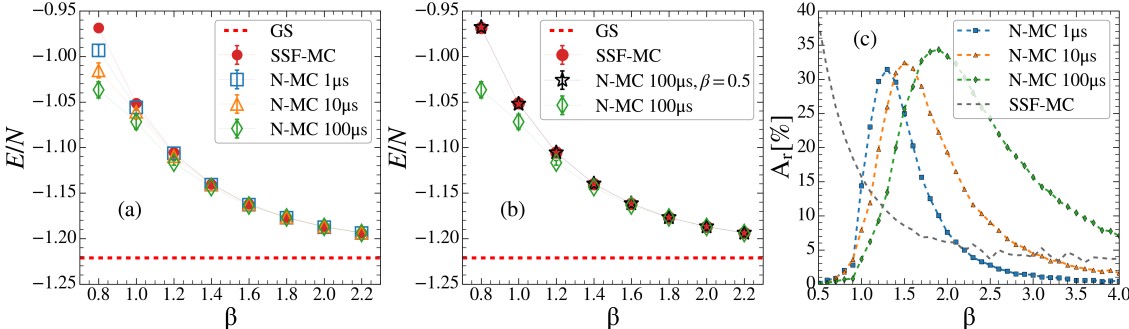

Figure 3: Panel (a): Average energy per spin $E/N$ as a function of the inverse temperature $\beta$, for the $N = 100$ lattice. The SSF-MC simulations (full red circles) are compared with three N-MC simulations driven by MADEs trained with different annealing times. The horizontal (red) dashed line indicates the ground-state (GS) energy. Panel (b): $E/N$ versus $\beta$ for the SSF-MC simulations (full red circles), the N-MC simulations with annealing time $t_a = 100\mu s$ (green empty rhombi), and the N-MC simulations corresponding to hybrid training data, including the QA configurations and SSF-MC simulations at $\beta = 0.5$ (blue empty stars). Panel (c): MH acceptance rates $A_r$ as a function of inverse temperature $\beta$. The SSF data (gray dashed curve) are compared to three N-MC datasets corresponding to different annealing times.

function $c(\tau)$, defined as

$$c(\tau) := \frac{\langle H_{t+\tau} H_t \rangle - \langle H_t \rangle^2}{\langle H_t H_t \rangle - \langle H_t \rangle^2}, \tag{16}$$

where the integers $t$ and $\tau$ count MC sweeps, $H_t$ is the energy of the configuration at sweep $t$, and the angular brackets indicate the average over the MC samples, discarding the thermalization regime. The definition of sweep for each algorithm is provided and motivated in Section 3. The role of the annealing times on the acceptance rates is also discussed below.

## 6.1   $N = 100$ **lattice**

The $N = 100$ spin glass is sufficiently small to be amenable to standard SSF-MC simulations, even in the low-temperature regime $\beta \simeq 1$. In Fig. 3, panel (a), we show the average energy per spin $E/N$ provided by N-MC simulations run for $\tau = 10^5$ sweeps. Three sets of simulations are performed, driven by NN trained with three annealing times. While at low temperatures all of them precisely agree with the (ground truth) SSF-MC results, significant deviations occur at higher $T$. The deviations are more sizable for the longer annealing times. We attribute these discrepancies to the lack of higher-energy samples in the D-Wave configurations, in particular for longer annealing times (see Fig. 2). Henceforth, the NN never samples high-energy configurations, while these have sizable Boltzmann weight at high $T$. This leads to an effective lack of ergodicity in the considered simulation times.

The lack of high-energy samples can be easily remediated considering a hybrid training dataset, including, e.g., $5 \cdot 10^4$ D-Wave configurations and just as many classical configurations. The latter are generated via a SSF-MC simulation performed at the relatively high temperature $\beta = 0.5$. As shown in panel (b) of Fig. 3, this data augmentation completely eliminates the bias in the N-MC predictions. The D-Wave configurations allow the NN learning how to sample low energies, while the classical configurations teach how to sample higher energies. This effect is further illustrate in panel (c), where we compare the acceptance rates of SSF-MC simulations with those of N-MC simulations based on D-Wave data. As expected, the former drop in the challenging low $T$ regime, while the N-MC updates become particular effective in

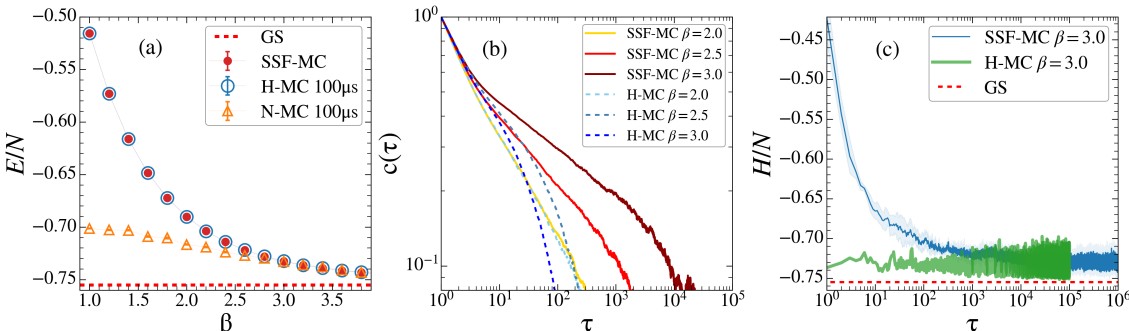

Figure 4: Panel (a): Average energy per spin $E/N$ as a function of the inverse temperature $\beta$, for the $N = 484\,(z = 4)$ lattice. The SSF-MC simulations (full red circles) are compared with an N-MC simulation (orange empty triangles) and with a H-MC simulations (blue empty circles). The horizontal (red) dashed line indicates the ground-state (GS) energy. Panel (b): Energy auto-correlation function $c(\tau)$ as a function of the number of sweeps $\tau$. The SSF-MC results at three inverse temperatures (dashed curves) are compared with the corresponding H-MC results. Panel (c): Configuration energy $H/N$ as a function of the number of sweeps $\tau$. An SSF-MC simulation at $\beta = 3$ (blue curve with shadow) is compared with the corresponding H-MC result (thick green curve). The semi-transparent shadow represents the fluctuations among 5 SSF-MC simulations.

that regime. This observation leads us to introduce the H-MC algorithm, which combines the two types of updates, as discussed in the next subsection. The H-MC algorithm circumvents the burden of creating the classical-configuration dataset. It is also worth noticing that the N-MC acceptance rates peak are lower temperatures for longer annealing times. This confirms that slow annealing allows the D-Wave configurations more accurately mimicking the low-temperature Boltzmann distribution.

## 6.2 $N = 484\,(z = 4)$ lattice

The larger lattice setup, including $N = 484\,(z = 4)$ spins, allows better observing glassy features in the $\beta \gtrsim 1$ regime. The SSF-MC simulations are here barely practical, requiring $\sim 10^8$ sweeps for reliable estimations of $E/N$ in the glassy regime. In Fig. 4, panel (a), we compare these predictions with H-MC results. The latter are obtained with only $4 \cdot 10^5$ sweeps, indicating a computation-time reduction by almost three orders of magnitudes. The agreement is precise. The correlation functions $c(\tau)$ corresponding to the SSF-MC and the H-MC algorithms are compared in panel (b). In the regime $\beta \gtrsim 2$, the H-MC algorithm outperforms the SSF algorithm, displaying orders of magnitude shorter correlation times. The performance boost is noticeable also in the thermalization process, visualized in panel (c) for the $\beta = 3$ case. The SSF-MC simulation equilibrates only after $\sim 10^5$ sweeps, while the H-MC equilibration time is negligible.

## 6.3 $N = 484\,(z = 8)$ lattice

Including also next nearest-neighbor diagonal couplings, corresponding to lattice connectivity $z = 8$ (for inner spins), provides an even more challenging computational testbed. For $\beta \gtrsim 2.5$, SSF-MC simulations performed with as many as $8 \cdot 10^7$ sweeps fail to ergodically explore the configuration space, leading to biased $E/N$ estimations. This is shown in the panel (a) of Fig. 5. A reliable efficiency benchmark is represented by the PT algorithm. Its predictions,

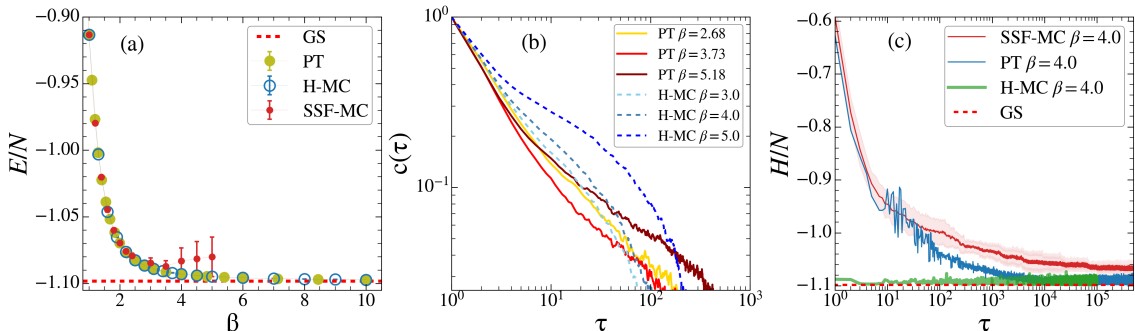

Figure 5: Panel (a): Average energy per spin $E/N$ as a function of the inverse temperature $\beta$, for the $N = 484\,(z = 8)$ lattice. The PT results (full yellow circles) are compared with H-MC simulations (empty blue circles) and with the average of 5 SSF-MC simulations run for $8 \cdot 10^7$ sweeps. The corresponding error-bars represent the estimated standard deviation of the mean of the 5 simulations. The horizontal (red) dashed line indicates the ground-state (GS) energy. Panel (b): Energy autocorrelation function $c(\tau)$ as a function of the number of sweeps $\tau$. The PT results at three inverse temperatures (dashed curves) are compared with the H-MC results at similar temperatures. Panel (c): Configuration energy $H/N$ as a function of the number of sweeps $\tau$. A PT simulation at $\beta = 4$ (blue curve) is compared with the corresponding H-MC result (thick green curve), and with the average of 5 SSF-MC simulations (red curve with shadow).

obtained with $5 \cdot 10^5$ sweeps performed after ex-post parameters optimization (see Section 3), are found to precisely agree with the H-MC results obtained with $4 \cdot 10^5$ sweeps. Notably, the agreement extends to extremely low temperatures $\beta \simeq 10$, where the energy expectation value $E/N$ almost coincides with the ground-state energy. Still, H-MC provides a significant benefit: while the PT simulation equilibrates only after $\sim 10^4$ sweeps, the H-MC displays negligible equilibration times.

The agreement between H-MC and PT simulations is further established by the energy histograms shown in panel (a) of Fig. 6 for the $\beta = 10$ case. In particular, the zoom on the low-energy region (see panel (b) of Fig. 6 and Table 2) demonstrates that the H-MC algorithm frequently samples very low energies, in particular the ground-state energy level, even when these energies are included neither in the D-Wave training data nor in the $4 \cdot 10^5$ configurations generated by the MADE (used as proposals). This indicates that the SSF updates allow the H-MC algorithm exploring relevant regions outside the reach of the MADE. Still, the neural updates suppress correlation times by performing large leaps in the configuration space.

## 7 Conclusions

While QAs are typically employed to tackle combinatorial optimization problems, we have described how to exploit them to boost the efficiency of thermodynamic-equilibrium simulations of Ising models. This is achieved via autoregressive generative NNs. These are trained on QA-generated data, and then used to drive the MC simulation. The augmentation of QA data with spin configurations generated by standard MC simulations has been explored. This allows extending the regime of applicability of the purely neural MC algorithm. Chiefly, a hybrid algorithm has been implemented. It exploits both neural proposals and standard SSF updates. It allows performing efficient ergodic simulations for challenging frustrated spin-glass models, both at high and at low temperatures, even approaching the ground-state energy. The neural

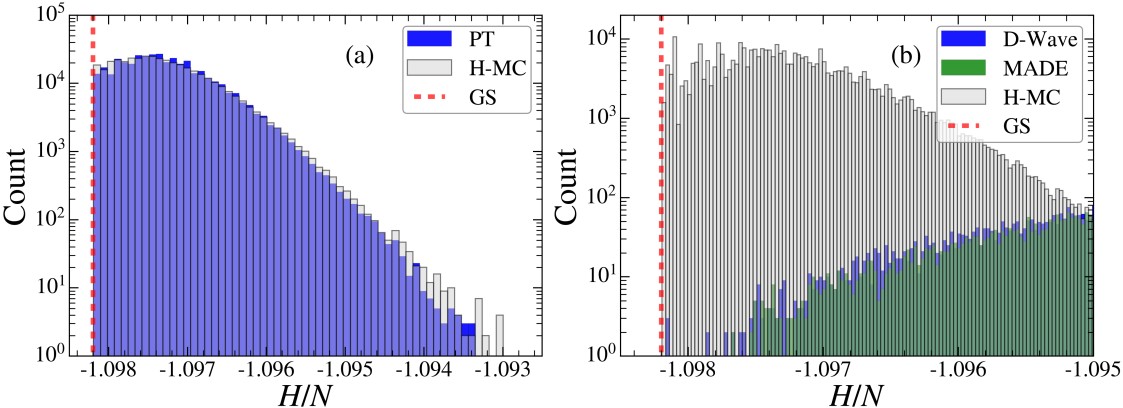

Figure 6: Panel (a): Histograms of $4 \cdot 10^5$ configuration energies per spin $H/N$, sampled by a PT simulation (blue) and by a H-MC simulation (gray with black contour) at $\beta = 10$, for the $N = 484$ ($z = 8$) lattice. The vertical (red) dashed line indicates the ground-state energy. Panel (b): Low-energy zoom on the histograms of $4 \cdot 10^5$ configuration energies per spin $H/N$, sampled in a H-MC simulations at $\beta = 10$ (gray with black contour), by a D-Wave QA with annealing time $t_a = 100\mu$s (blue), and by the trained MADE (green).

updates allow performing large leaps in configuration space with sufficient acceptance rates. The standard SSF proposals allow exploring the neighborhoods of the configurations reached by the neural proposals, thus exploring otherwise inaccessible regions. The hybrid algorithm outperforms standard SSF simulations, and it is competitive with PT, but with the significant benefit of a much faster equilibration.

The effect of generating QA configurations with different annealing times $t_a$ has been analyzed. Even for relatively short annealing times, these are found to be sufficiently representative of the relevant low-energy configurations to provide a speed-up in neural and hybrid MC simulations. While it has been argued that the samples from the D-Wave QAs might follow a Boltzmann distribution at an effective temperature [68,69], our neural and hybrid approaches do not assume this is the case, meaning that the training configurations might follow a different distribution. The MH acceptance stage and the combination with SSF updates anyway allow us sampling the Boltzmann distribution at the desired temperature without bias.

Future endeavors should focus on further exploring the role of the annealing time in order to optimize the usage of QA time. Auto-correlation functions corresponding to different observables could be analyzed. The neural cluster updates of Ref. [22] might be introduced to compensate the expected diminishing of acceptance rates for larger systems. Adaptive MC schemes featuring on-the-fly learning [17] might also be helpful. Furthermore, protocols to directly generate proposals from the QA, as recently shown in the case of gate-based quantum computers [37], might be explored.

**Code and datasets**    To favor future comparative studies, we provide via the Zenodo repository our datasets [70], including the couplings $J_{ij}$, the D-Wave QA configurations, the energy expectation values $E/N$, as well as the codes [71] for training the MADE and for running all MC algorithms discussed in this article.

# Acknowledgments

Interesting discussions with G. Mazzola, R. Fazio, and G. E. Santoro are acknowledged. We acknowledge the Cineca award under the ISCRA initiative, for providing access to D-Wave quantum computing resources, and PRACE for awarding access to the Fenix Infrastructure resources at Cineca, which are partially funded by the European Union's Horizon 2020 research and innovation program through the ICEI project under the Grant Agreement No. 800858. This work was partially supported by the Italian Ministry of University and Research under the PRIN2017 project CEnTraL 20172H2SC4.

# A   Optimal intra-chain coupling strength

The lattice setups we consider (see Section 3) cannot always be directly implemented on the Pegasus graph of the D-Wave Advantage QA. The D-Wave interface uses a heuristic embedding procedure to assign each logical spin variable to one or to more physical qubits of the device [72]. In the latter case, we have a chain of qubits with a strong nearest-neighbor ferromagnetic coupling $J_c$.

The corresponding Hamiltonian term reads: $\hat{H}_{chain} = -J_c \sum_i \sum_{\langle k,k' \rangle}^{n_i} \hat{\sigma}^z_{i,k} \hat{\sigma}^z_{i,k'}$, where $\hat{\sigma}^z_{i,k}$ is a Pauli matrix at qubit $k$ of chain $i$, and $n_i$ is the chain length. This term is introduced to force the $n_i$ qubits to act a single variable.

While the D-Wave interface provides reasonably effective default values for $J_c$, manual tuning allows users optimizing the QA performance, meaning that the sampling of low-energy configurations is boosted. Indeed, weak couplings allow the qubits of the same chain to decouple, therefore breaking the correspondence with the problem Hamiltonian. In such cases the spin readout is based on majority voting [73]. Excessive intra-chain couplings induce clustering phenomena, detrimental for the annealing dynamics [74]. The optimal intra-chain coupling strength also depends on the typical interaction strengths among logical qubits.

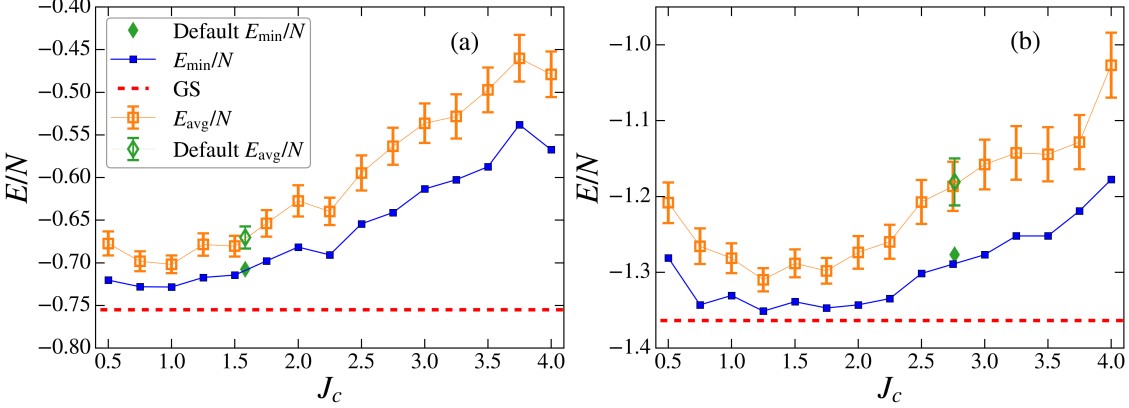

Figure 7: Average energy per spin $E_{avg}/N$ (orange empty squares) and corresponding minimum $E_{min}/N$ (blue full squares) of $10^3$ configurations sampled by a D-Wave QA, as a function of the intra-chain ferromagnetic coupling $J_c$. The (green) empty and full rhombi correspond to the average and minimum obtained with the default coupling of the D-Wave interface, respectively. Panel (a): the couplings $J_{ij}$ are sampled from Unif$[-1, 1]$. Panel (b): the couplings $J_{ij} = \pm 1$ are sampled from binary random distribution.

Two exemplary optimizations are visualized in Fig. 7, for the $N = 484\,(z = 4)$ lattice setup

and for the annealing time $t_a = 10\mu s$. One notices that reducing the intra-chain coupling compared to the default values allows both the mean and the minimum energies approaching the exact ground-state value. This effect is more pronounced for the uniform random couplings $J_{ij} \sim \text{Unif}[-1, 1]$ [panel (a)], compared with, e.g., binary random couplings $J_{ij} = \pm 1$ [panel (b)]. In fact, the latter case appears to represent a less challenging optimization problem, given that the minimum energy almost reaches the ground state when the optimal intra-chain coupling is set.

## B   D-Wave total run time

It is worth mentioning that the actual utilization time of the D-Wave QA extends beyond the annealing time $t_a$ per sample. For the D-Wave Advantage system, the required time $T$ for one call to the D-Wave interface is computed as:

$$T = t_p + N_s(t_a + t_r + t_d), \tag{17}$$

where $t_p$ is the programming time, $t_r$ is the readout time per sample, $t_d$ is the delay time between two consecutive readouts per sample, and $N_s$ is the number of requested configurations. Since the allowed call time $T$ is limited, so is the number of configurations that can be sampled in one system call. For the considered lattices, the number of configurations $N_s$ in a call ranges from $10^3$ to $10^4$, depending on the problem size and the chosen annealing time. To generate larger datasets, several system calls are performed and, to ensure consistency, all QA parameters are fixed and the same embedding map is used. For example, for 100 samples of the $N = 484\,(z = 8)$ lattice setup, with annealing time $t_a = 100\mu s$, a total of 0.150 seconds of D-Wave QA time is used, with $t_p \simeq 15\text{ms}$, $t_r \simeq 110\mu s$, and $t_d \simeq 995\mu s$.

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
