# Peer review of "Accelerating equilibrium spin-glass simulations using quantum annealers via generative deep learning"

_SciPost Physics_

## Round 1 · Referee Report · Anonymous (Referee 1) · 2023-3-8

Report

This is a report on "Accelerating equilibrium spin-glass simulations using quantum annealers via generative deep learning" by Scriva and collaborators.

The Authors present a thorough demonstration of how configurations generated by a quantum annealer may be used to train neural networks, to provide an alternative to classical monte carlo sampling of equilibrium correlations. The paper is clearly written and convincingly substantiates a link between different research areas of equilibrium statistical mechanics and quantum adiabatic machines. It fulfills the minimum criteria for publication in SciPost.

As a non-expert in machine learning and neural networks I appreciated the clarity of the manuscript, and also the clear explanation of the logic of training with experimental data. On the physics side of things -- it was not clear how/if finite physical temperature of the annealer enters the analysis. The authors touched on this issue in the paragraph towards the end where they discuss the interplay with finite annealing time. Perhaps, they can clarify the issue further, if/when they revise the manuscript.
  • validity: -
  • significance: -
  • originality: -
  • clarity: -
  • formatting: -
  • grammar: -

Author:  Giuseppe Scriva  on 2023-04-28  [id 3629]

(in reply to Report 1 on 2023-03-08)
Category:
answer to question
reply to objection
pointer to related literature

**Our reply:**

We thank the Referee for the positive assessment.
We agree with them that a clearer discussion on the possible effect of the physical temperature of the device is required. In the revised manuscript, we better describe the role on the annealing time on the energies sampled by the quantum annealer. While the MC acceptance rates do peak at different temperatures for different annealing times, and this might indeed lead one to define an effective temperature, this temperature is not related to the device temperature but rather reflects the effects of too fast annealing protocols. These comments are bow included in an extended discussion, together with more detailed references, in the Conclusions section.

---

## Round 1 · Referee Report · Anonymous (Referee 2) · 2023-4-4

Strengths

  1. The manuscript introduces a clear and neat application of quantum annealers, which is of great interest.

  2. It promotes and justifies further research in application of quantum annealers to produce samples to train machine learning codes.

Weaknesses

  1. Numerical evidence is sufficient but not massive.

Report

In this paper the authors report on a study on how to accelerate Monte Carlo simulations with a generative neural network that selects smart proposals which helps decide swap updates during the execution of the algorithm. The crucial point is that, to this end, one requires sampling a probability distribution of spin glass models which can be done efficiently and more quickly using a D-wave computer, for low temperatures. For high temperatures the authors conclude it has to be done use single spin-flip and neural proposals. The data produced is used to train the generative neural network and optimise the algorithm. The result outperforms the single spin-flip Metropolis-Hastings algorithm and competes with parallel tempering. The paper represents a clever and promising application of quantum annealers and I recommend publication. A few comments are the following. The numerical evidence is sufficient in my opinion but not massive. Some improvement is obtained in some cases, but there is a vast variety of parameters that can be tuned or explored, for example effect of different activation functions, more variety in lattice sizes, etc. Also, comparison is made with some algorithms, like single-spin flip or parallel tempering. There are more proposals in the recent literature. Anyhow, in my opinion, this work introduces a good enough indication that sampling with a quantum annealer is promising and justifies publication.

Requested changes

  1. I would like to ask to include in the conclusions section a small report on other recent strategies and algorithms, further than parallel tempering.

  • validity: top
  • significance: top
  • originality: top
  • clarity: high
  • formatting: perfect
  • grammar: perfect

Author:  Giuseppe Scriva  on 2023-04-28  [id 3628]

(in reply to Report 2 on 2023-04-04)
Category:
answer to question
reply to objection

**Our Reply:**

We thank the Referee for the positive assessment of our manuscript.
We agree with them that a short report on recent research on improved MC algorithms is appropriate. This is included in the "Conclusions" section of the revised manuscript. It briefly discusses algorithms such as, e.g., non-reversible MC and population annealing, beyond others. This report also allowed us to emphasize that our hybrid scheme can be combined with any of these improved algorithms. Notice that the additional References [70-82] have been included.

---

## Editorial Decision

resubmitted